# Adipocyte-specific deletion of Dbc1 does not recapitulate healthy obesity phenotype but suggests regulation of inflammation signaling

Leonardo Santos[1], Rafael Sebastián Fort[2,3], Geraldine Schlapp[4], Karina Cal[1], Valentina Perez-Torrado[1], Maria Noel Meikle[4], Ana Paula Mulet[4], Camila Espasandín[1,5], Camila Chiesa[1], José R. Sotelo-Silveira[2,6], Jose M. Verdes[7], Paola Contreras[8], Aldo J. Calliari[1,5], Martina Crispo[4], Jose L. Badano[9], Carlos Escande[1]*

1 Laboratory of Metabolic Diseases and Aging, Institut Pasteur de Montevideo, Montevideo, Uruguay, 2 Departamento de Genómica, Instituto de Investigaciones Biológicas Clemente Estable, Montevideo, Uruguay, 3 Sección Genómica Funcional, Facultad de Ciencias, Universidad de la República, Montevideo, Uruguay, 4 Unidad de Biotecnología en Animales de Laboratorio, Institut Pasteur de Montevideo, Montevideo, Uruguay, 5 Department of Biosciences, Facultad de Veterinaria, Universidad de la República, Universidad de la República, Montevideo, Uruguay, 6 Departamento de Biología Celular y Molecular, Facultad de Ciencias, Universidad de la República, Montevideo, Uruguay, 7 Unidad Patología, Departamento de Patobiología, Facultad de Veterinaria, Universidad de la República, Montevideo, Uruguay, 8 Departamento de Fisiología, Facultad de Medicina, Udelar, Uruguay, 9 Human Molecular Genetics Laboratory, Institut Pasteur de Montevideo, Montevideo, Uruguay

* escande@pasteur.edu.uy

## Abstract

The protein Deleted in Breast Cancer 1 (Dbc1) is an important regulator of various transcription factors and epigenetic modulators, significantly influencing metabolism, obesity, and aging-related processes. Knockout mice lacking Dbc1 exhibit severe obesity but remain protected from liver steatosis, insulin resistance, and atherosclerosis. We hypothesized that this phenotype of "healthy obesity" results from adipose tissue expansion, which prevents free fatty acid spillover and subsequent metabolic damage to peripheral tissues. To further investigate the putative role of Dbc1 in adipose cells during obesity and its effects on metabolic dysregulation, we generated conditional Dbc1 knockout (KO) mice by backcrossing with AdipoQ-CRE transgenic mice to selectively abrogate Dbc1 expression in all mature adipocytes (Dbc1$^{LoxP/LoxP}$;CRE). These mice demonstrated effective deletion of Dbc1 in mature adipocytes across various fat depots. We assessed the impact of Dbc1 deletion on metabolic regulation in male and female mice fed standard chow and high-fat diets. Our findings revealed that Dbc1 knockout in mature adipocytes did not influence weight gain, glucose tolerance, or other metabolic dysregulation markers, irrespective of sex. However, Dbc1 KO adipocytes exhibited an mRNA expression profile indicative of heightened inflammation during obesity. These results suggest that the protective phenotype observed in whole-body Dbc1 KO obese mice is not attributable to Dbc1's function within mature adipocytes but likely involves other cell types in adipose

**Data availability statement:** The raw FASTQ data sets supporting the results of this article are available at the Sequence Read Archive repository (https://www.ncbi.nlm.nih.gov/sra) (Accession number SRR30574701).

**Funding:** This study was supported by multiple funding sources, including Agencia Nacional de Investigación e Innovación (Award Number: FCE_1_2014_1_104002, Recipient: Carlos Escande), FOCEM (Award Number: COF 03/11, Recipient: Carlos Escande), CSIC-DT program (Recipient: Aldo J. Calliari), Ministerio de Educación y Cultura (UY) (Award Number: FVF_2023_504, Recipient: Leonardo Santos), CSIC Uruguay (Recipient: Aldo J. Calliari), and PEDECIBA Udelar (Recipient: Leonardo Santos). The funders had no role in study design, data collection and analysis, decision to publish, or preparation of the manuscript.

**Competing interests:** The authors have declared that no competing interests exist.

tissue. Moreover, the specific deletion of Dbc1 in mature adipocytes unveils a novel role of Dbc1 in inflammation signaling during obesity.

## Introduction

Obesity and its related metabolic dysfunctions are a leading cause of morbimortality worldwide. While the irruption of GLP1 agonists has shown that treatment of obesity and allied comorbidities is no longer utopian, several aspects of the molecular signaling that determine the onset and progression of obesity-related metabolic dysfunction remain elusive. Accumulating evidence shows that the functional status of adipose tissue plays a determinant role in the onset and progression of systemic metabolic dysfunction during obesity. Adipose tissue inflammation, hypoxia and fatty-acid spillover are among the main determinants that affect metabolic dysfunction and determine the fate of several other organs, such as the liver, pancreas and skeletal muscle (revised in [1]).

We and others have previously shown that the protein Deleted in Breast Cancer 1 (Dbc1) plays a key role in the onset and progression of metabolic dysfunction during obesity. We found that Dbc1 KO mice develop morbid obesity characterized by a healthy expansion of adipose tissue. Importantly, Dbc1 KO mice were protected against fat tissue inflammation and cellular senescence in adipose tissue [2,3], fatty liver disease, and atherosclerosis [3].

Dbc1 was shown to bind and regulate several target proteins that are key players in metabolic regulation. Among them are transcription factors (p53, Foxp3, Rev-erbα), epigenetic modifiers, such as SIRT1, HDAC3 and SUV39H1, DNA repair enzymes (BRCA1 and PARP1) and the nuclear receptors ER-α and ER-β. Interestingly, by binding and activating IKK-β at the cytoplasm, Dbc1 promotes NFkB-dependent transcriptional activation of various genes involved in inflammation and other immune responses[4–10].

Such promiscuity in its binding and regulation capacity makes understanding the specific roles of Dbc1 a challenging issue. Indeed, the specific role of Dbc1 seems to depend on the cell type and context. In an effort to understand the exact function of Dbc1 in regulating adipose tissue function during obesity, we engaged in generating a conditional, tissue-specific knockout model. We generated this mouse model by inserting LoxP sites the *Dbc1* gene using CRISPR/Cas9 technology. After obtaining the modified mouse, we backcrossed it with adipocyte-specific CRE transgenic mice (Tg(AdipoQ-CRE)1Evdr) in order to specifically delete Dbc1 in mature adipocytes. Adipocyte-specific Dbc1 KO males and females were characterized under normal chow and high-fat diets. We found that adipocyte-specific KO mice did not recapitulate our original findings in Dbc1 whole-body KO mice. Both males and females gained similar weight and showed comparable results for all markers tested compared to control animals. Finally, we tested if deletion of Dbc1 in mature adipocytes had any functional consequence on gene expression profiles during obesity. RNAseq data from adipocytes isolated from obese control and conditional Dbc1 KO

mice showed a molecular fingerprint of inflammation in those animals in which Dbc1 was deleted. The latter suggests that the healthy phenotype of the whole body Dbc1 KO mouse, attributed to protection against adipose tissue inflammation, is independent of Dbc1's role in mature adipocytes. Alternatively, it may involve other cells present in the tissue. We conclude that in mature adipocytes, Dbc1 may still be playing a role in regulating inflammatory signaling during obesity, but the exact mechanism to achieve the so-called healthy phenotype seems to be different from that originally proposed.

## Materials and methods

### Reagents and antibodies

Unless otherwise specified, all reagents and chemicals were purchased from Sigma-Aldrich. Anti-rabbit DBC1 antibody was purchased from Bethyl Laboratories (Cat. # A300-434A), and mouse monoclonal anti-B-actin was from Sigma-Aldrich (Cat. # A2228).

### Animal handling and experiments

All mice used in this study were bred and maintained at the Institut Pasteur Montevideo Animal facility (UBAL). The experimental protocol was approved by the Institutional Animal Care and Use Committee of the Institut Pasteur Montevideo (CEUA; protocol numbers 70153-000839-17, 003-19, and 006-19). All studies described were performed according to the methods approved in the protocol and following all international guidelines and national legal regulations (Law 18.611). Mice received a standard chow or high-fat diet (42% fat and 0.25% cholesterol, AIN93G, LabDiet, USA) and water ad libitum. Weight gain was monitored during obesity experiments in Dbc1$^{LoxP/LoxP}$;CRE (n = 24 females, n = 18 males) and in control littermates Dbc1$^{LoxP/LoxP}$ (n = 30 females, n = 17 males). In chow diet experiments, weight gain was assessed in Dbc1$^{LoxP/LoxP}$ (n = 7 females, n = 6 males) and Dbc1$^{LoxP/LoxP}$;CRE (n = 6 females, n = 9 males).

Tolfenamic acid was administered subcutaneously (1 mg/kg, Tolfedine, Vetoquinol, Madrid, Spain) in order to provide analgesia and anti-inflammatory effects, when necessary. At the end of the experiment, animals were sacrificed under deep anesthesia (a mixture of ketamine (150 mg/kg, Pharmaservice, Ripoll Vet, Montevideo, Uruguay) and xylazine (15 mg/kg, Seton 2%; Calier, Montevideo, Uruguay), followed by cervical dislocation.

### Generation of the Dbc1 conditional knock-out mice

All animal procedures to generate the mutant line were performed at the SPF animal facility of the Laboratory Animal Biotechnology Unit of Institut Pasteur de Montevideo. Experimental protocols were opportunely approved by the Institutional Animal Ethics Committee (protocol number 007–18), in accordance with National Law 18.611 and international animal care guidelines (Guide for the Care and Use of Laboratory Animal) [11] regarding laboratory animal protocols. Mice were housed in individually ventilated cages (Tecniplast, Milan, Italy) containing chip bedding (Toplit 6, SAFE, Augy, France), in a controlled environment at 20 ± 1°C with a relative humidity of 40–60%, in a 14/10 h light-dark cycle. Autoclaved food (Labdiet 5K67, PMI Nutrition, IN, USA) and autoclaved filtered water were administered *ad libitum*.

Cytoplasmic microinjection was performed on B6D2F2 or C57BL/6J zygotes using a mix of 100 ng/μl Cas9 mRNA, 25 ng/μl of each sgRNA, and 50 ng/μl ssDNA oligo diluted in microinjection buffer. Viable embryos were cultured overnight in M16 medium microdrops under embryo-tested mineral oil, containing the ligase IV inhibitor SCR7, in 5% $CO_2$ in air at 37°C. The next day, 2-cell embryos were transferred into the oviduct of B6D2F1 0.5 days post coitum (dpc) pseudopregnant females (an average of 20 embryos/female), following surgical procedures [12]. For surgery, recipient females were anesthetized with a mixture of ketamine (100 mg/kg) and xylazine (10 mg/kg). Tolfenamic acid was administered subcutaneously (1 mg/kg, Tolfedine, Vetoquinol, Madrid, Spain) in order to provide analgesia and anti-inflammatory effects [13]. Pregnancy diagnosis was determined by visual inspection by an experienced animal caretaker two weeks after embryo transfer, and litter size was recorded on day 7 after birth.

Pups were tail-biopsied and genotyped 21 days after birth, and mutant animals were maintained as founders. The first round of microinjections resulted in the insertion of only the 3' LoxP site. *Dbc1* 3' LoxP F1 animals were used to produce embryos by *in vitro* fertilization using the Center for Animal Resources and Development (CARD) protocol [14]. Three hours after fertilization, zygotes were microinjected into the cytoplasm with CRISPR reagents at the same aforementioned concentrations, to insert the lacking 5' LoxP site. Viable two-cell embryos were transferred to pseudopregnant females as detailed above. Founders with both 3' and 5' LoxP sites were used to generate the final mutant line.

The mice used in this study were backcrossed with C57BL/6J mice to homogenize the genetic background. They were then bred to homozygosity for the floxed Dbc1 allele. To generate adipocyte-specific Dbc1 knockout mice, the homozygous floxed Dbc1 mice were crossed with B6.FVB-Tg(AdipoQ-CRE)1Evdr/J mice obtained from Jackson Laboratories.

The experimental colony was established by crossing *AdipoQ-CRE* Dbc1$^{LoxP/LoxP}$ hemizygous mice with Dbc1$^{LoxP/LoxP}$ mice. For the experiments, groups with a sample size (N) of ten or more mice per genotype were used.

## Blood glucose measurements

Mice were fasted for 16 hours prior to fasting glucose measurements and glucose tolerance tests (GTT). For the GTT, mice received an intraperitoneal injection of a glucose solution at a dose of 1.5 g/kg of body weight. Plasma glucose concentrations were measured from tail blood using a handheld glucometer (Accu-Chek, Roche). Under the normal diet (ND) condition, six Dbc1$^{LoxP/LoxP}$ and six Dbc1$^{LoxP/LoxP}$;CRE mice were used for each sex. In the high-fat diet (HFD) condition, the male groups consisted of seven Dbc1$^{LoxP/LoxP}$ and nine Dbc1$^{LoxP/LoxP}$;CRE mice, while the female groups included six Dbc1$^{LoxP/LoxP}$ and six Dbc1$^{LoxP/LoxP}$;CRE mice. Area under the curve (AUC) values were calculated for the GTT, and comparisons were performed using unpaired t-tests.

## Hepatic and renal function

To assess liver and kidney function, whole blood from Dbc1$^{LoxP/LoxP}$ (n = 5 males, n = 8 females) and from Dbc1$^{LoxP/LoxP}$;CRE (n = 10 males, n = 8 females) mice was analyzed using the Pointcare V2 Chemistry Analyzer (Tianjin MNCHIP Technologies Co., China). The parameters determined were total proteins (TP), albumin (ALB) level, globulin (GLO) level, ALB/GLO ratio, total bilirubin (BIL T) level, alanine aminotransferase (ALT) level, aspartate aminotransferase (AST) level, gamma-glutamyl transpeptidase (GGT) level, blood urea nitrogen (BUN) level, creatinine (CRE) level, BUN/CRE ratio, and glucose (GLU) level.

## Isolation of WAT adipocytes

Fat depots were dissected and weighed. A collagenase digestion solution (collagenase type II, GIBCO 1 mg/ml in Hank's Balanced Salt Solution containing 2% BSA) was added at a 3:1 ratio of collagenase solution to fat tissue, and the tissue was finely minced and incubated for 1 hour at 37°C in a shaking incubator. Then, the adipocyte suspension was centrifuged at 100g for 10 minutes at room temperature (RT). Subsequently, floating mature adipocytes were washed twice with cold PBS.

## Western blotting

Adipocytes were lysed using radioimmunoprecipitation assay buffer (at a 1:10 volume ratio) supplemented with 5 mM NaF, 5 mM nicotinamide, 50 mM β-glycerophosphate, 1 μM trichostatin A (catalog no.: 647925; Sigma), a protease inhibitor cocktail, and then sonicated in five cycles of 10 seconds each, followed by intervals of 5 seconds. Homogenates were incubated for 20–30 minutes at 4 °C under constant agitation and then centrifuged at 10,000g for 10 minutes. Protein concentration in the supernatants were determined using the Bradford protein assay reagent. Samples were resuspended in a Laemmli buffer, separated by SDS-PAGE, and transferred to polyvinylidene fluoride membranes. After blocking

(Tris-buffered saline containing 0.2% Tween-20 and 5% nonfat milk), the membranes were incubated overnight with the appropriate antibodies. Secondary antibodies were incubated for 1 hour and detected using SuperSignal West Pico Chemiluminescent Kit (catalog no.: 34080; Pierce).

## Histology

Liver sections were obtained from paraffin-embedded tissue and stained with hematoxylin and eosin (H&E) following standard procedures. For each mouse, at least three sections were analyzed. In both the normal diet (ND) and high-fat diet (HFD) groups, three Dbc1LoxP/LoxP and three Dbc1LoxP/LoxP;CRE mice were used.

## Plasma NEFA quantification

In males, 8 Dbc1LoxP/LoxP and 9 Dbc1LoxP/LoxP;CRE mice were used in the high-fat diet (HFD) group, while 5 Dbc1LoxP/LoxP and 5 Dbc1LoxP/LoxP;CRE mice were used in the normal diet (ND) group. In females, 6 Dbc1LoxP/LoxP and 6 Dbc1LoxP/LoxP;CRE mice were used in both ND and HFD groups. Non-esterified fatty acids were measured using the NEFA-HR Assay (Wako) according to the manufacturer's instructions. Statistical analysis was performed using an unpaired t-test.

## Measurements of lipid accumulation in the liver

Liver triglycerides were measured according to [15]. Liver samples from Dbc1LoxP/LoxP (n = 5 males, n = 5 females) and Dbc1LoxP/LoxP;CRE (n = 5 males, n = 5 females) were homogenized in lysis buffer, as described above. Lipid content was measured using the TG Color GPO/PAP AA kit (Wiener Lab, USA) following the manufacturer's instructions.

## Adipocyte Lysis and RNA Extraction

Total RNA from adipocytes was extracted following lysis with TRIzol reagent (Thermo Fisher Scientific) according to the manufacturer's instructions. Briefly, adipocytes were harvested and resuspended in 1 mL of TRIzol reagent. The cells were lysed by pipetting up and down several times and then incubated at room temperature for 5 minutes to ensure complete dissociation of nucleoprotein complexes. Following lysis, 0.2 mL of chloroform was added per 1 mL of Trizol reagent. The mixture was vigorously shaken for 15 seconds and then incubated at room temperature for 2–3 minutes. The samples were centrifuged at 12,000 x g for 15 minutes at 4°C. The aqueous phase was then processed using the Direct-zol RNA Miniprep Kit (Zymo Research) according to the manufacturer's protocol.

## RNA quality control, library and sequencing

An Infinite 200 PRO (Tecan) spectrophotometer was used to evaluate the concentration and the 260/280 ratio of the RNA obtained from mature adipocytes. The RNA quality was evaluated using the RNA integrity number (RIN), determined with a Bioanalyzer RNA 6000 Nano chip (Agilent, Cat# 5067–1511). All samples demonstrating high RNA integrity, with an average RIN of 9.0, were used for further studies. RNA-seq was performed on samples from individual animals (n ≥ 3) of each experimental group. All libraries were prepared and sequenced by Genewiz (https://www.genewiz.com/) using the TruSeq mRNA Sample Prep Kit v2 protocol (poly(A)+ selection) and Illumina HiSeq 2x150 bp sequencing..

## RNA-seq analysis

Raw reads (FASTQ files) obtained from Genewiz (https://www.genewiz.com/) were used as the input for miARma-Seq pipeline, a comprehensive tool for miRNA and mRNA analysis [16], following the user manual (v 1.7.2). Low quality reads and adapter sequences were removed with Cutadapt software [17], allowing a minimum read Phred quality score of 20. Filtered high-quality reads (Ave. 54,995,205 ± SD 6,123,659 reads, S1 Table in S1 File) were aligned to Mus musculus reference genome (GRCm38/mm10, indexed from http://bowtie-bio.sourceforge.net/bowtie2/index.shtml) using HISAT2

[18] with default parameters (Overall genome mapping %: Ave. 94.9 ± SD 0.3, S1 Table in S1 File). Gene counts were assessed with featureCounts [19] using default parameters (S1 Table in S1 File). For quantification, we use the annotation coordinates of Ensembl Mus_musculus.GRCm38.96 GTF file. Normalization and Differential Expression Genes (DEGs) analysis were conducted using SARTools pipeline [20] in RStudio, selecting the edgeR algorithm [21], Trimmed Mean of the M-values (TMM) normalization, and CPM (Counts Per Million) ≥ 1 as the cut-off (S1 Table and S4 Fig in S1 File). A significance threshold of log2 fold change > |0.58| and p-value <0.05 was applied for DEGs, instead of the adjusted p-value, in order to have a sufficient number of genes to perform over-representation analysis.

### RNA-seq validation by qPCR

Expression levels of candidate DEGs involved in identified enriched pathways were confirmed by qPCR. Genes not directly involved in such pathways but significantly up or down regulated according to RNA-seq data, were also included. Tissues were homogenized in Trizol for RNA extraction according to the manufacturer's protocol. DNase I treatment was used to eliminate genomic DNA contamination. Reverse transcription was done using SuperScript II RT, and quantitative reverse transcription polymerase chain reaction (qRT-PCR) was performed using a Fast SYBR Green. Relative quantification of changes in gene expression were expressed in relation to housekeeping genes. Expression was calculated as fold increase with respect to control. The primers were from IDT and their sequences are provided in S2 Table in S1 File.

### Gene set enrichment analysis

Gene Set Enrichment Analysis (GSEA) was conducted using the gseGO function from the clusterProfiler R package (v4.8.2) [22]. This analysis examined Gene Ontology (GO) terms related to biological processes by considering the complete list of genes, ranked by their fold change in expression, rather than focusing solely on differentially expressed genes (DEGs). The terms with significance threshold of adjusted p-value <0.05 were selected.

### Statistics

Unless stated otherwise, values are shown as the mean ± standard deviation (SD) from three to five experiments. Statistical analyses were conducted using GraphPad Prism for most experiments, while RNA-seq data were analyzed using R (version 3.6) with the enrichplot, xlsx, ggplot2, and heatmap.2 libraries (employing Euclidean distance and Ward clustering algorithms). Group differences were evaluated using either ANOVA or a two-tailed Student's t-test, and a p-value below 0.05 was considered statistically significant.

## Results

### Generation of Dbc1^LoxP/LoxP conditional mice by CRISPR/Cas9 and adipocyte-specific deletion

The mouse *Dbc1* gene, located on chromosome 14, comprises 21 exons forming an open reading frame that extends from exon 2 to exon 21. It is predicted to have only one full-length transcript encoding a protein of 922 amino acids. To create a conditional knockout (cKO) murine line, we targeted exon 3 to be floxed. We designed two guide RNAs (gRNAs) to target intron 2 and intron 3 using CRISPR/Cas9 and assessed their effectiveness by transiently transfecting a murine NIH3T3 cell line that stably expresses Cas9, followed by heteroduplex analysis (Fig S1 in S1 File).

Additionally, we generated two single-stranded DNA (ssDNA) donors for knock-in of LoxP sites. These sequences comprise part of intron 2 and intron 3, LoxP sites, and two restriction enzyme sites, which allow for the identification of the insertion. In the first step, we inserted the 3' LoxP site. A second round of genetic manipulation was necessary to insert the 5' LoxP site.

We crossed a male mouse carrying the selected mutation with C57BL/6J females to establish the colony and performed five rounds of backcrossing before beginning the characterization of the line. This process should be sufficient to

segregate any off-target or unwanted modifications of the genome produced during genetic manipulation. From that point onward, we maintained the line by crossing homozygous mutant animals. To achieve knockout in mature adipocytes, we crossed the floxed *Dbc1* animals with the *AdipoQ-CRE* mouse line (Fig 1A).

First, we sequenced the region where the genetic manipulation was performed to confirm the expected sequence. We used primers located near the 5' and 3' ends of exon 3 of Dbc1 to sequence the PCR product. The expected mutations were detected, and no unwanted mutations were found (Fig 1B). Next, to verify the construct, we performed PCR on genomic DNA from cKO MEF cells transiently expressing Cre recombinase. The presence of a 300 bp PCR product confirmed that DNA recombination occurred successfully (Fig 1C).

Next, we evaluated Dbc1 at the protein level by western blot. The ablation of exon 3 (300 bp) by Cre recombinase was expected to cause a shift in the open reading frame of the transcript and the introduction of a premature stop codon. The mutant allele could therefore produce a 35-amino acid peptide. In agreement with that prediction, the ~100 kDa Dbc1 band was absent in adipocytes from different fat tissue depots. In addition, in whole tissue extracts from white adipose tissue (WAT) and brown adipose tissue (BAT), Dbc1 protein levels decreased. In contrast, there was no difference in protein expression in the liver and brain (Fig 1D). Although an apparent increase in Dbc1 expression was observed in the isolated stromal vascular fraction (SVF) from CRE + fat tissue (Fig 1D), its quantification using replicates (N = 4), revealed no statistically significant differences (S1 Fig in S1 File).

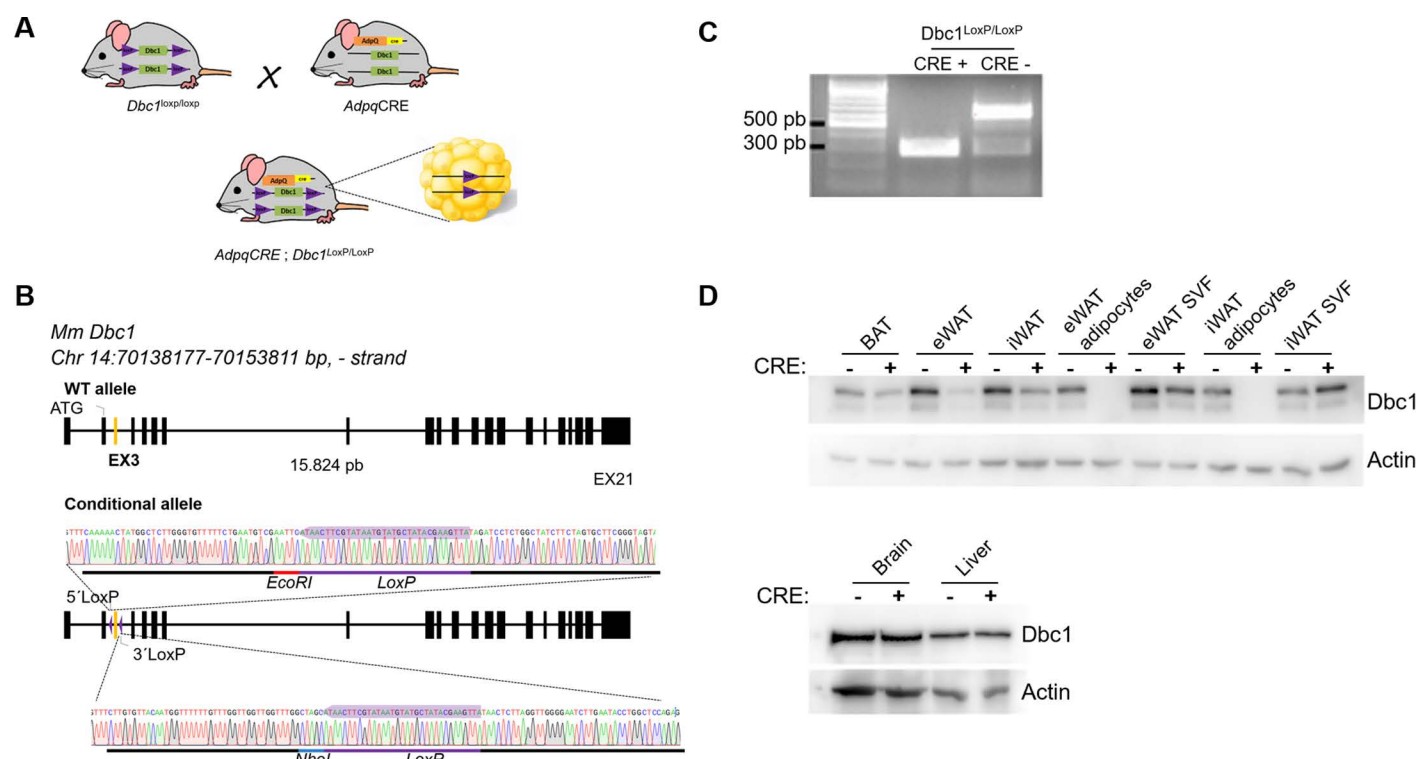

**Fig 1. Generation of an Adipocyte-Specific Dbc1 Knockout Mouse Model. (A)** Breeding strategy used to generate adipocyte-specific Dbc1 knockout mice. The combination of the *AdipoQ-CRE* transgene with Dbc1 floxed alleles led to the deletion of Dbc1 specifically in adipocytes. **(B)** Schematic representation of Dbc1 gene targeting, showing LoxP sites flanking exon 3. **(C)** Genotyping by PCR confirming recombination at the Dbc1 floxed locus in adipocytes from mice carrying both the Adipo*Q-CRE* transgene and Dbc1 floxed alleles. **(D)** Western blot analysis of DBC1 protein expression in different adipose tissue depots (inguinal, iWAT; epididymal, eWAT), isolated adipocytes, stromal vascular fraction (SVF), brain, and liver.

## Metabolic characterization of Dbc1$^{LoxP/LoxP}$;CRE KO male and female mice on a normal diet

In previous studies, we and others showed that whole-body Dbc1 knockout mice do not develop obesity when fed a regular chow diet [15,23]. However, they present abnormal glucose management that is mainly due to altered liver gluconeogenesis [23]. This effect was a consequence of dysregulation of hepatic REV-ERBα activity and increased PEPCK expression [18,23]. Consistently, the specific deletion of Dbc1 only in mature adipocytes, did not affect body weight or glucose homeostasis, measured by GTT in normal chow conditions, in both male and female mice (Fig 2). Free-fatty acid levels were also comparable among genotypes in both males and females (Fig 2). In conclusion, the absence of Dbc1 specifically in mature adipocytes failed to influence body weight, glucose management, or free-fatty acid levels in mice under normal chow conditions.

## Metabolic characterization of Dbc1$^{LoxP/LoxP}$;CRE KO male and female mice on high-fat diet

Next, we evaluated the effect of diet-induced obesity on the metabolic phenotype of Dbc1$^{LoxP/LoxP}$;CRE KO mice. Our previous work, conducted in mice with a mixed genetic background, showed that whole-body Dbc1 KO mice fed with high-fat

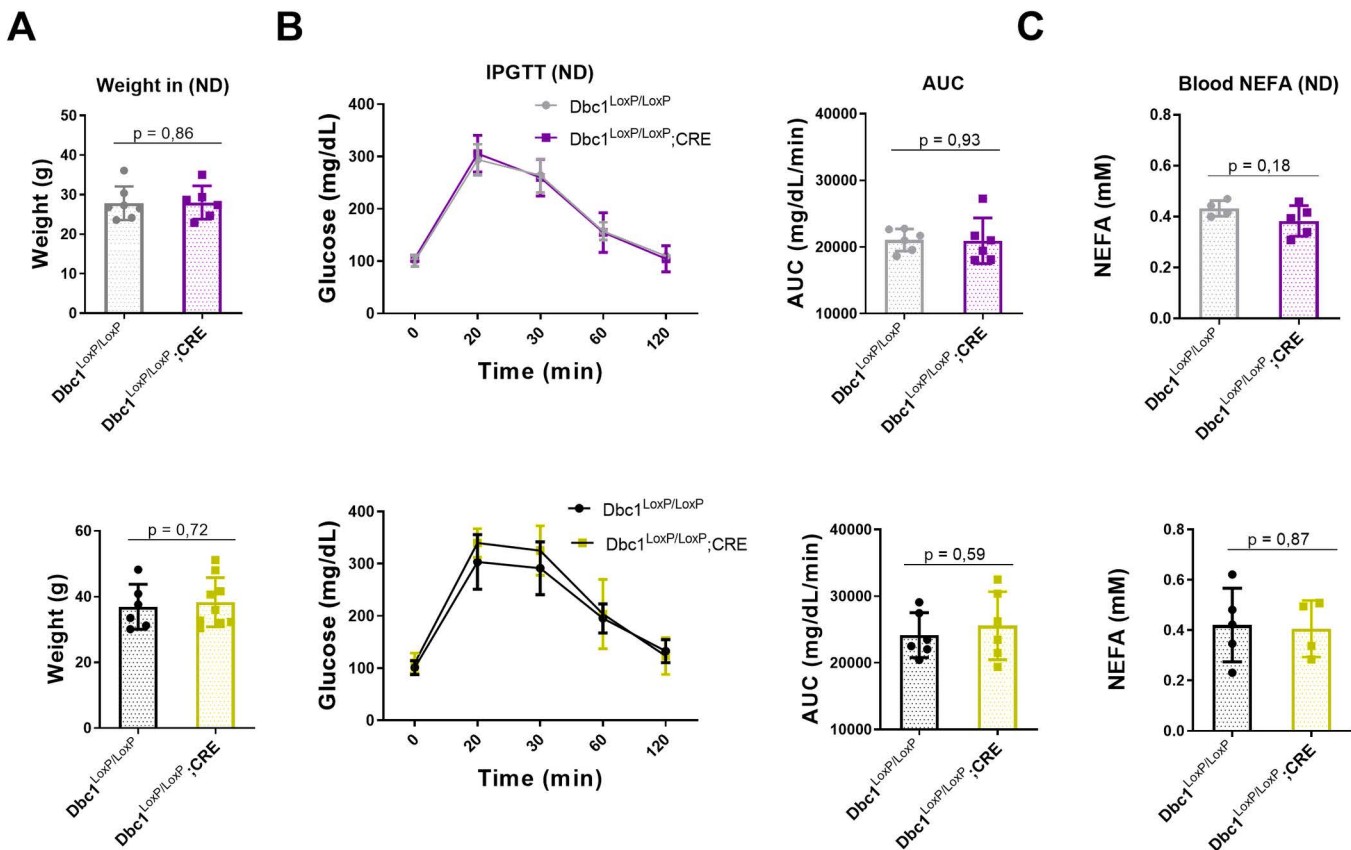

**Fig 2. Similar Weight, Glucose, and Blood Free-Fatty Acid Levels Across Different Genotypes under a Normal Diet (ND). (A)** Mean body weight of female (gray and purple) and male mice (black and yellow) from different genotypes. **(B)** Glucose Tolerance Test (GTT). Glucose response evaluation in male and female mice from different genotypes. **(C)** Blood Free-Fatty Acid Levels of male and female mice from different genotypes. Data represent the mean ± standard deviation (SD) for each group. Statistical analysis was performed using a t-test, and no statistically significant differences were observed.

diet for 20 weeks, develop morbid obesity with prevention of fatty-acid spillover and metabolic damage [3]. This phenotype was also observed in mice with pure C57BL/6J genetic background (S2 Fig in S1 File).

When Dbc1$^{LoxP/LoxP}$;CRE mice were fed a high-fat diet (HFD), both males and females gained weight at a rate similar to that of their control littermates (Fig 3A). Weight gain was monitored for up to 20 weeks on HFD, and no significant differences were observed between genotypes. Furthermore, we did not detect any differences in glucose tolerance (Fig 3B) or in plasma free fatty acid levels (Fig 3C), which are key indicators of the healthy obesity phenotype we previously described [3].

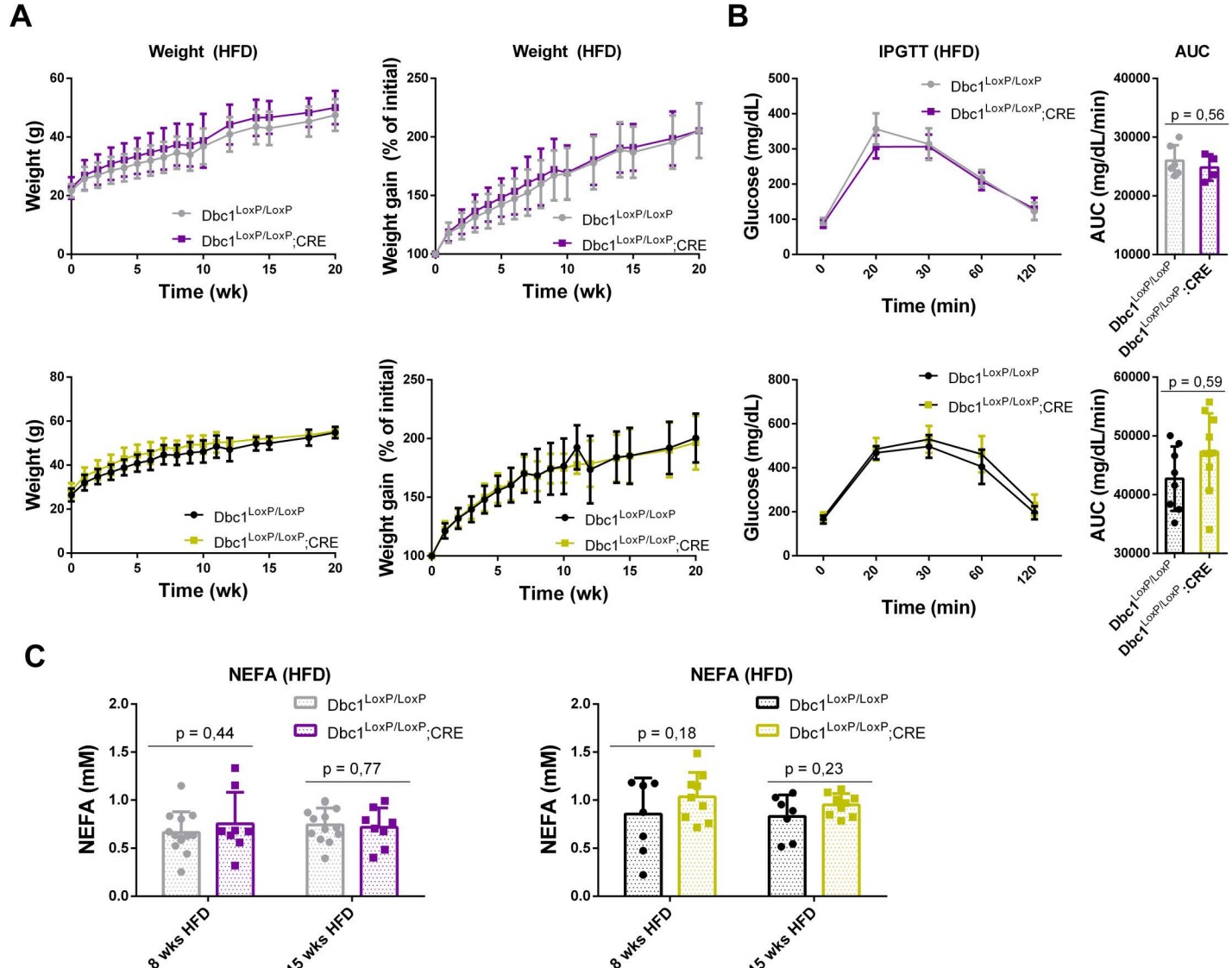

**Fig 3. Development of Diet-Induced Obesity (DIO) in the Dbc1$^{LoxP/LoxP}$;CRE mouse model.** **(A)** Comparable weight gain was observed across both sexes and genotypes during HFD administration, with no significant differences detected. **(B)** Glucose tolerance test was assessed after 8 weeks on the HFD regimen. No significant differences were found in glucose tolerance between the experimental groups. **(C)** Plasma Free-Fatty Acid Levels of serum samples collected after 8 and 15 weeks of HFD administration were analyzed for free fatty acid concentrations.

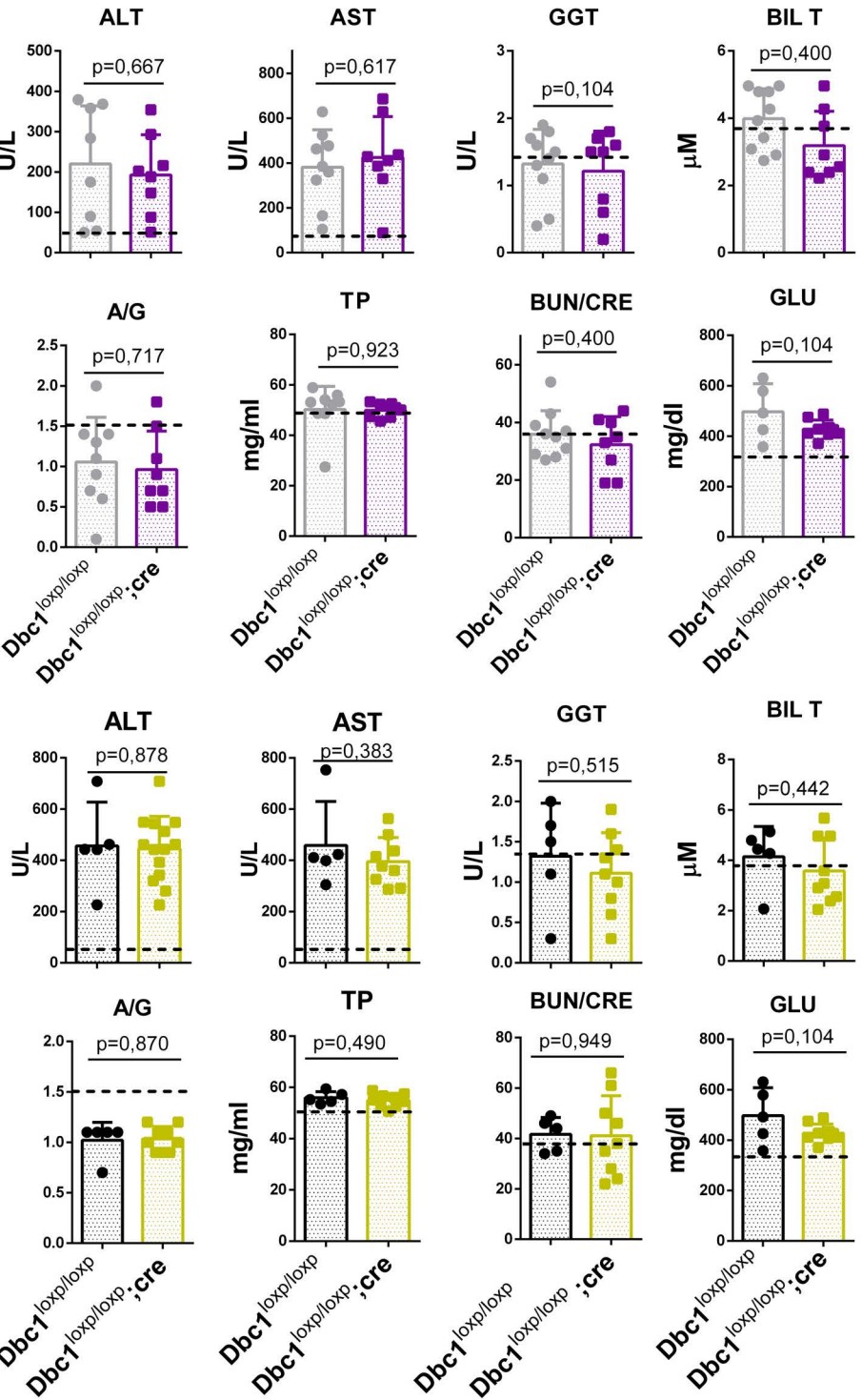

**Fig 4. Renal and Hepatic Function in Male and Female Mice of Both Genotypes.** Graphs illustrating liver and kidney function data obtained from blood analysis of female (gray and purple) and male mice (black and yellow) following 15 weeks of high-fat diet (HFD) administration. The data represent the evaluation of renal and hepatic markers, including total bilirubin (BIL T), BUN/CRE, total protein (TP), albumin and globulin levels (ALB and GLOB), aspartate aminotransferase (AST), gamma-glutamyl transferase (GGT), alanine transaminase (ALT), albumin-to-globulin ratio (A/G), and non-fasting glucose (GLU). The dashed line shows biomarker levels from normal chow-fed mice. Results are shown for both genotypes. Data represent the mean ± standard deviation (SD) for each group. Statistical analysis was performed using a t-test.

We measured several markers of liver and renal function after 15 weeks of a high-fat diet, in both males and females. Despite a clear effect of diet on the levels of several markers, we found no differences among genotypes (Fig 4). Kidney function markers, such as the albumin to globulin ratio (A/G) and blood urea nitrogen to creatinine ratio (BUN/CRE), indicated that renal function was normal compared to levels in chow-fed mice. Hepatic enzymes alanine aminotransferase (ALT) and aspartate aminotransferase (AST) showed a significant increase in blood levels compared to lean mice, indicating an effect of the high-fat diet on hepatic physiology. However, bilirubin levels and gamma-glutamyl transferase (GGT) activity remained unchanged relative to controls, suggesting that liver damage was moderate. Under HFD, both genotypes accumulated triglycerides in the liver without significant differences.

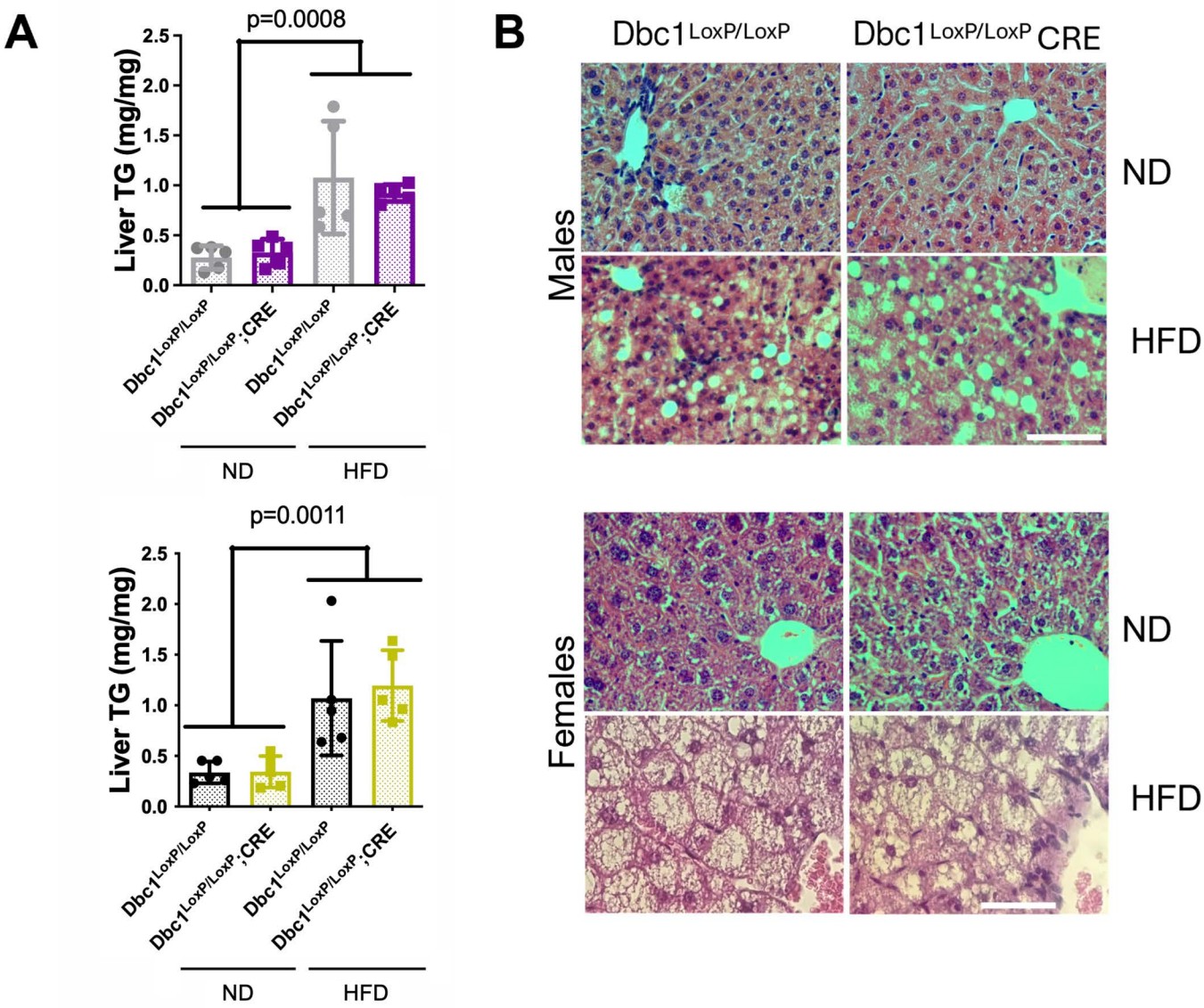

**Fig 5. Hepatic Responses to Diet-induced Obesity (A)** Liver triglyceride content in both sexes and genotypes under Normal Diet and High-Fat Diet (HFD). Data represent mean±standard deviation (SD). Statistical analysis was performed using two-way ANOVA. **(B)** Hematoxylin and eosin staining of liver sections from male mice under Normal Diet and High-Fat Diet. Scale Bar: 50 μm.

Histological analysis of liver samples confirmed the presence of steatosis, as expected in the diet-induced obesity (DIO) condition (Fig 5). Overall, Dbc1<sup>LoxP/LoxP</sup>;CRE KO mice did not show significant differences in weight gain, glucose management, or liver and kidney function compared to controls under a high-fat diet treatment, indicating that adipocyte-specific Dbc1 deletion does not replicate the protective effects observed in whole-body Dbc1 knockout models.

### RNA-seq analysis of isolated white adipocytes in WT and Dbc1<sup>LoxP/LoxP</sup>;CRE KO obese mice

In an effort to determine the role Dbc1 is playing in mature adipocytes during obesity, we performed RNA-seq analysis of isolated adipocytes from Dbc1<sup>LoxP/LoxP</sup>;CRE mice and their control littermates after 20 weeks of high-fat diet. Whole-transcriptome sequencing was conducted on adipocytes isolated from five Dbc1<sup>LoxP/LoxP</sup>;CRE knockout mice and three control mice. We performed transcriptome analyses to identify differentially expressed genes (DEGs) and functional pathways using bioinformatics approaches. RNA-Seq revealed 60 DEGs with $\log_2 FC|0.58|$ and p-value < 0.05 between control and Dbc1 KO adipocytes (Fig 6A). Among these, 42 genes were upregulated in Dbc1 KO adipocytes, while 18 genes were downregulated in control adipocytes, with a 1.5-fold change (Fig 6B). Overall, the data indicate that Dbc1 has a moderate influence on the transcriptional landscape of adipocytes during diet-induced obesity, potentially affecting their function. To explore the functions of the identified DEGs, we analyzed which pathways and cellular processes were enriched for each comparison using Enrichr (https://maayanlab.cloud/Enrichr/) and the Reactome 2022 database [24]. We found significant pathways enriched only for the upregulated DEGs, with the top three pathways being related to immune system response: Immune System, Cytokine Signaling in Immune System, and Signaling of Interleukins (Fig 6C). To surpass the constraints of over-representation analysis based on a limited number of DEGs, we employed Gene Set Enrichment Analysis (GSEA) involving the entire set of genes in the transcriptome. This is a well-known strategy that permits exploring the complete list of genes ranked based on the FC of expression between conditions [22]. We conducted this analysis for Gene Ontology terms using the ClusterProfiler algorithm. This approach identified significantly enriched Gene Ontology terms (Biological Process). According to the GO dataset, Dbc1 knockout (KO) adipocytes showed global upregulation in several inflammatory response pathways (Fig 6D). Particularly noticeable are the positive enriched terms related with cytokines production and Nf-κb transcriptional activity (Fig 6E).

In addition, selected genes identified in the RNA-seq analysis were validated by qPCR, showing changes that were consistent to those observed in the transcriptomic data. Moreover, qPCR was performed to assess inflammatory markers such as Il1β and Tgfβ. Although these markers did not display significant differences in the RNA-seq analysis of isolated adipocytes, their expression in whole adipose tissue suggested a trend toward increased inflammation in the knockout mice. These qPCR data are shown in S5 Fig in F1 File. Globally, this analysis underscores the involvement of Dbc1 in regulating adipocyte inflammatory responses.

## Discussion

As a regulator of several transcription factors and epigenetic modulators, Dbc1 has emerged as a multifaceted determinant in metabolic regulation. Interestingly, Dbc1 knockout mice develop morbid obesity but are protected against liver steatosis, insulin resistance, and atherosclerosis. It has been proposed that this expansion of the adipose tissue avoids free-fatty acid spillover and metabolic damage in peripheral tissues, thus accounting for the observed healthy phenotype. In order to further understand the roles of Dbc1 in adipose tissue and its associated phenotype, we generated a conditional knockout mouse model aimed to specifically delete Dbc1 in mature adipocytes.

The generation of Dbc1 conditional KO mice was successful, confirmed by the correct insertion of LoxP sites and the absence of unwanted mutations. Adipocyte-specific deletion of Dbc1 was validated through PCR and Western blot, showing the ablation of the Dbc1 protein in adipocytes from Dbc1<sup>LoxP/LoxP</sup>;CRE KO animals. Western blot analysis of whole adipose tissue samples revealed that, in addition to adipocytes, the stromal vascular fraction also expresses high levels of the protein. This precise genetic manipulation enabled us to isolate the role of Dbc1 specifically in adipocytes, independent of its functions in other tissues.

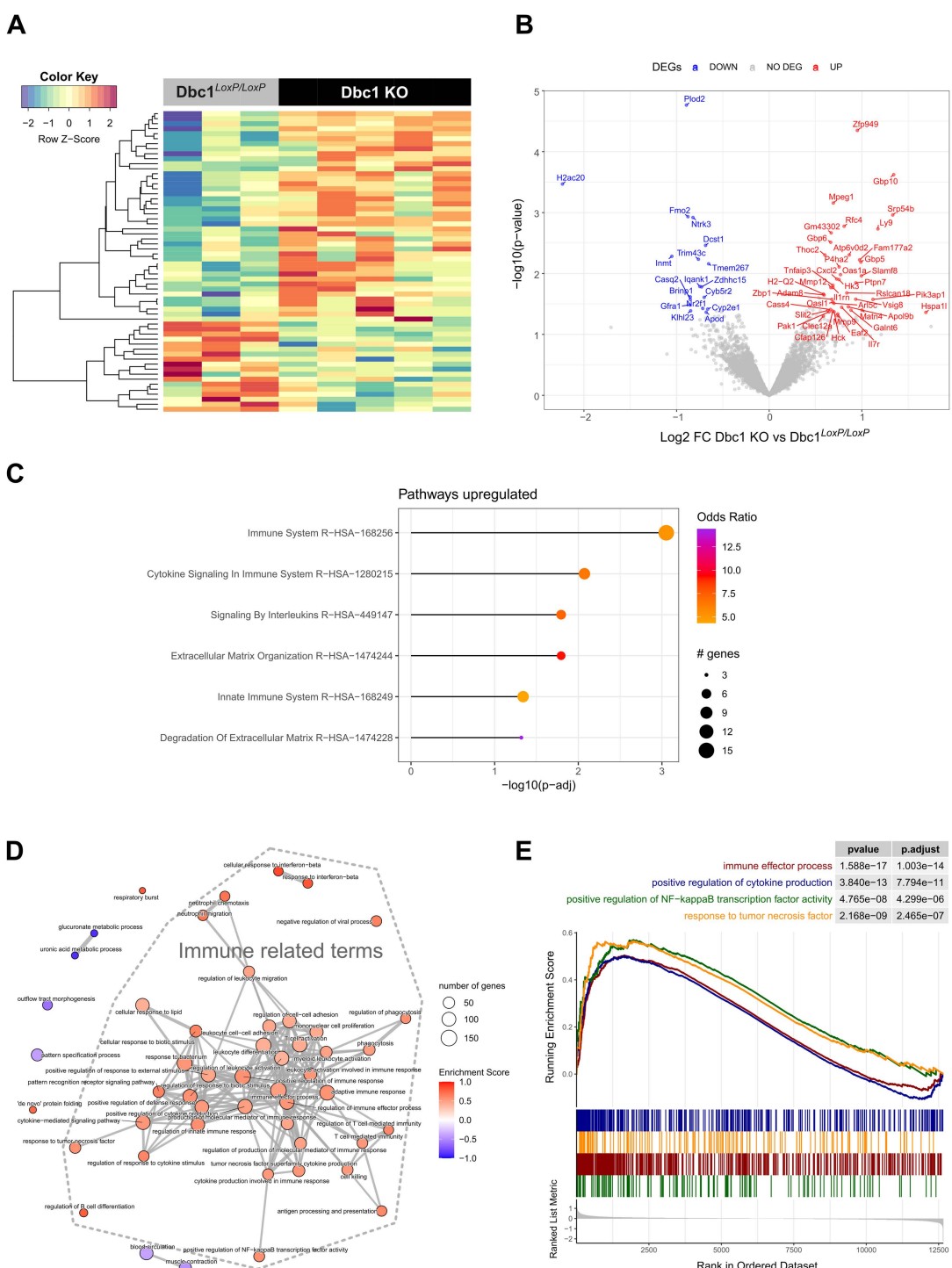

**Fig 6. Transcriptomic data of mature adipocytes from Dbc1 KO vs WT mice** **(A)** Heatmap of differentially expressed protein coding genes (DEGs) with log$_2$FC|0.58| and p-value<0.05. **(B)** Volcano plot highlighting upregulated (red) and downregulated (blue) genes. **(C)** Reactome-enriched pathways identified for upregulated DEGs. **(D)** Emapplot of the top 50 significantly enriched Gene Ontology terms (Biological Process), with increased terms in red and decreased terms in blue, in Dbc1 KO compared to Dbc1$^{LoxP/LoxP}$. **(E)** Gene Set Enrichment Analysis (GSEA) of selected Gene Ontology terms from **(C)**. The x-axis represents the ranked genes based on log2 (Fold Change); the y-axis (upper panel) shows the Enrichment Score value, while the y-axis (bottom panel) represents the value log$_2$FC of the ranking metric.

Our metabolic characterization of Dbc1$^{LoxP/LoxP}$;CRE KO mice under normal diet conditions showed no significant differences in body weight, glucose tolerance, or free-fatty acid levels compared to control mice. Under high-fat diet conditions, both male and female Dbc1$^{LoxP/LoxP}$;CRE KO mice gained weight similarly to their control littermates, with no significant differences in glucose management, free-fatty acid levels, or markers of liver and renal function. These observations contrast with previous studies on whole-body Dbc1 knockout mice, which exhibited altered glucose management due to liver-specific mechanisms.

The failure to recapitulate the whole-body Dbc1 knockout phenotype in Dbc1$^{LoxP/LoxP}$;CRE KO mice indicates that Dbc1 expression in adipocytes is not responsible for the overall metabolic health or susceptibility to diet-induced obesity. Instead, Dbc1 expression in adipocytes might play a more subtle or context-dependent role, a possibility that requires further investigation under specific stress or metabolic conditions. As a whole, these findings put forward that the metabolic protection observed in whole-body Dbc1 knockout models may arise from non-adipocyte cells located into or even outside the adipose tissue. Moreover, the previously observed healthy obesity phenotype could be a result of complex interactions between different organs rather than being specific to a particular tissue or cell type. Further investigation is needed to clarify these results.

Our RNA-Seq analysis provided new insights into the molecular functions of Dbc1 in adipocytes. We identified several inflammatory pathways that are overrepresented in Dbc1 KO adipocytes. Notably, there is evidence of an inflammatory repressor role for Dbc1 in lymphocytes [25]. In fact, Dbc1 selectively suppresses alternative NF-κB pathways, which is in line with our GSEA analysis (S3 Fig in S1 File). These findings seem to be paradoxical, and intriguing given the diminished inflammation observed in the adipose tissue of whole-body Dbc1 KO mice [3]. Interestingly, it has been reported that adipocyte inflammatory pathways, such as TNF signaling, are important for proper adipogenesis [26]. Therefore, it could be hypothesized that a reduced ability to sense and respond to proinflammatory stimuli at the level of the adipocyte leads to a decrease in the capacity for healthy adipose tissue expansion and remodeling. Thus, the elevated immune response in Dbc1 KO adipocytes observed in this study could be biologically relevant for the healthy expansion of adipose tissue seen in the whole-body KO model.

The upregulation of inflammatory pathways observed in Dbc1 knockout adipocytes points to a complex role for Dbc1 in maintaining adipocyte homeostasis and makes it difficult to provide a simplistic explanation that reconciles with the diminished adipose tissue inflammation previously observed in whole-body Dbc1 KO mice [3]. Clearly, other cellular components of adipose tissue cooperate in maintaining adipose tissue homeostasis. The mechanistic links between Dbc1 and these inflammatory processes warrant further exploration, particularly to understand whether Dbc1 directly interacts with key signaling molecules involved in these pathways.

Our study challenges the belief that adipocyte-expressed Dbc1 protects against liver steatosis, insulin resistance, and atherosclerosis, instead revealing a nuanced role of Dbc1 in regulating gene expression and inflammatory responses in adipocytes, particularly under diet-induced obesity. Despite the lack of significant metabolic alterations following adipocyte-specific Dbc1 deletion, the observed transcriptional changes suggest that Dbc1 may be relevant for adipocyte function during obesity.

## Supporting information

**S1 File. Fig S1.** Testing of gRNA Efficiency in Inducible 3T3-Cas9 Cells. (A) Western blot analysis of total cell extracts at various time points following incubation with doxycycline. We generated a 3T3-L1 cell line where Cas9 expression is controlled by doxycycline. (B) Immunofluorescence images of cells after 48 hours of incubation with doxycycline. Nuclei are stained with DAPI, and Flag-Cas9 is detected by immunofluorescence following doxycycline induction. Bar = 50 μm. (C) Acrylamide gel electrophoresis stained with EtBr showing PCR products using genomic DNA from 3T3-Cas9 cells after transfection with various gRNAs and treatment with doxycycline. Black arrows indicate the formation of heteroduplexes, which are absent in the controls of uninduced and untransfected cells, as well as in induced but untransfected cells

(Ctrl and Ctrl2, respectively). A noticeable decrease in amplicon yield suggests significant modifications following Cas9 cleavage and DNA repair processes. (D) Western blot analysis of DBC1 protein levels in the stromal vascular fraction of adipose tissue from Dbc1LoxP/LoxP and Dbc1LoxP/LoxP;CRE mice, showing DBC1 and Tubulin as a loading control. (E) Densitometric analysis of Dbc1 normalized to Tubulin and expressed as fold change relative to Dbc1LoxP/LoxP. Data are presented as mean±SD (n=4 per group). Statistical significance was determined using an unpaired t-test. **Fig S2.** Metabolic protection of Dbc1 KO obese mice. (A) Weight gain in WT and Dbc1 KO mice fed a high-fat diet on a C57BL/6J background, showing increased weight gain in Dbc1 KO mice compared to WT control mice. (B) Fasting glucose vs. body weight in obese mice shows a positive correlation between fasting glucose and body weight in WT (wild-type) animals, as expected. However, dbc1 knockout (KO) animals do not show this correlation, reflecting the protection against developing metabolic syndrome observed in Dbc1 KO animals. **Fig S3.** Pathway Analysis of Gene Expression Changes in Dbc1 KO adipocytes. The selected pathway, as diagrammed in KEGG, with gene expression changes (fold change, FC) projected based on the data. Genes marked in red are overexpressed in Dbc1 KO, while those in green are downregulated. Genes shown in gray indicate no change in expression between the two conditions. **Fig S4.** Pathway Analysis and processing of the RNA-seq samples. A Pearson correlationmatrix of the simples. Density plot of the simples pre B and post C filtering low read counts features. Boxplot of the simples pre D and post E normalization was applied. **Fig S5.** Validation of RNA-seq data by qPCR. (A) Gene expression levels of Il1b, Tgfb, Tnfa, and Nfkb were measured by qPCR in adipose tissue from Dbc1loxP/loxP and Dbc1loxP/loxP;CRE mice. Data are presented as mean±SD (n=6 per group) and normalized to Actb (β-actin). qPCR results support the trends observed in RNA-seq data Statistical significance was determined using unpaired t-test. (B) Validation of RNA-seq findings by selecting genes with low p-values and high fold changes. Expression levels of selected genes were assessed by qPCR, consistent with differential expression patterns detected in transcriptomic analysis.
(ZIP)

**S1 Table.**
(XLSX)

**S2 Table.**
(XLSX)

## Author contributions

**Conceptualization:** Jose L. Badano, Carlos Escande.

**Data curation:** Rafael Sebastian Fort, Camila Chiesa.

**Formal analysis:** Rafael Sebastian Fort, Carlos Escande.

**Funding acquisition:** Carlos Escande.

**Investigation:** Leonardo Santos, Rafael Sebastian Fort, Geraldine Schlapp, Karina Cal, Valentina Perez-Torrado, Maria Noel Meikle, Ana Paula Mulet, Camila Espasandín, Paola Contreras, Aldo J. Calliari.

**Methodology:** Leonardo Santos, Jose L. Badano.

**Project administration:** Carlos Escande.

**Resources:** Carlos Escande.

**Supervision:** José R. Sotelo-Silveira, Jose M Verdes, Martina Crispo, Carlos Escande.

**Validation:** Jose M Verdes, Carlos Escande.

**Writing – original draft:** Leonardo Santos, Carlos Escande.

**Writing – review & editing:** Leonardo Santos, Camila Chiesa, Aldo J. Calliari, Martina Crispo, Jose L. Badano.

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
