## [Decision Letter · Decision Letter 0]

21 Oct 2024

PONE-D-24-40711Adipocyte-specific deletion of Dbc1 does not recapitulate healthy obesity phenotype but suggests regulation of inflammation signalingPLOS ONE

Dear Dr. Escande,

Thank you for submitting your manuscript to PLOS ONE. After careful consideration, we feel that it has merit but does not fully meet PLOS ONE’s publication criteria as it currently stands. Therefore, we invite you to submit a revised version of the manuscript that addresses the points raised during the review process.

We look forward to receiving your revised manuscript.

Kind regards,

Chien-Feng Li, M.D., Ph.D.

Academic Editor

PLOS ONE

Journal Requirements:

2. To comply with PLOS ONE submissions requirements, in your Methods section, please provide additional information regarding the experiments involving animals and ensure you have included details on (1) methods of sacrifice, (2)  efforts to alleviate suffering.

“CE was supported by grants by ANII (FCE_1_2014_1_104002), CSIC, PEDECIBA and FOCEM (COF 03/11). AC was supported by the CSIC-DT program, from the Udelar. LS was supported by grant FVF_2023_504. This work was supported by FOCEM - Fondo para la Convergencia Estructural del Mercosur (COF 03/11).”

5. Please note that your Data Availability Statement is currently missing the DOI/accession number of each dataset or a direct link to access each database. If your manuscript is accepted for publication, you will be asked to provide these details on a very short timeline. We therefore suggest that you provide this information now, though we will not hold up the peer review process if you are unable.

**Additional Editor Comments:**

Dear authors,

Based on the review reports I am inviting you to revise your work as per the reviewers' comments.

Reviewers' comments:

Reviewer's Responses to Questions

**Comments to the Author**

1. Is the manuscript technically sound, and do the data support the conclusions?

Reviewer #1: Yes

Reviewer #2: Yes

2. Has the statistical analysis been performed appropriately and rigorously? 

Reviewer #1: No

Reviewer #2: Yes

3. Have the authors made all data underlying the findings in their manuscript fully available?

Reviewer #1: No

Reviewer #2: Yes

4. Is the manuscript presented in an intelligible fashion and written in standard English?

Reviewer #1: Yes

Reviewer #2: Yes

5. Review Comments to the Author

Reviewer #1: The authors present their findings from characterization of mature adipocyte specific knockout of Dbc1. Unexpectedly, no difference is observed in wildtype and cAT-dbc1 KO animals on normal chow or HFD except for enhanced immune signaling in KO adipocytes. The findings though new and important in context to the data on the whole body knockout mice, can be distilled into a short report rather than a primary research article. Most of the data on cAT-dbc1 KO mouse generation can be supplemental. RNA seq data on major immune genes should be validated by qPCR. Methods must clearly show the number of mice used in each experiment. There are some minor errors of grammar, line formatting, etc.

Reviewer #2: The transcription factor Deleted in Breast Cancer 1 (DBC1) has previously been shown to regulate obesity and aging-related processes in preclinical models, with murine global Dbc1 KO causing morbid obesity but protecting against liver steatosis, insulin resistance and atherosclerosis. This makes DBC1 a promising therapeutic target. In this study the authors generated an adipocyte-specific Dbc1 KO model to assess whether the putatively beneficial effects of global Dbc1 KO are attributable to adipocyte Dbc1 KO. Deletion of Dbc1 in mature adipocytes had no effect on weight gain, glucose tolerance nor other markers of metabolic dysregulation, regardless of sex, however. This is the central finding of this study, and although it is essentially a “negative” finding this is a clear and valuable answer to the question posed. Dbc1 KO adipocytes were, however, reported to display an mRNA expression profile consistent with increased inflammation, although the magnitude of this was not very impressive, and the finding was not pursued in more detail. In summary, this study adds new and useful information regarding the metabolic role of adipocyte Dbc1. In several places the drafting of the manuscript could also be included.

Major Comments

1. The manuscript states that “In previous studies, we and others showed that whole-body Dbc1 knockout mice do not develop obesity when fed a regular chow diet” (L291-293) but later states that “Our previous work, in mice with mixed genetic background, showed that whole-body Dbc1 KO mice develop morbid obesity with prevention of fatty-acid spillover and metabolic damage” (L314-316). Can this be reconciled? Presumably it relates to diet?

2. Fig 1: In Panel D, top, the immunoblot shows a clear decrease of Dbdc1 protein expression in Cre+ whole eWAT and iWAT and corresponding isolated adipocytes, suggesting successful deletion of Dbc1 in mature adipocytes. However, expression of Dbdc1 in isolated SVF seems increased compared to related whole fat tissue, particularly in the last lane, possibly indicating compensatory upregulation in SV cells from Cre+ fat tissue. Can the authors quantify the blot and clarify this?

3. Dbc1 knockout (KO) adipocytes show upregulation of several inflammatory pathways (Fig 6D). Particularly noticeable are the enriched terms related to cytokine production and NF-kB activity. This may indicate the adipose tissue is inflamed in KO mice, but this requires substantial further characterisation if much is to be made of this observation. As noted above, the negative metabolic findings are perhaps the most important outcome of the study.

Minor comments

1. The layouts of figure legends, and panel labelling are not very clear. As an example, in L267-289, the illustration jumps back and forward or repeats description of the same things.

2. Nomenclature for the mouse strain is sometimes inconsistent: cAT-Dbc1 KO vs Dbcloxp/loxp Cre +/- (L291- 301). Also, can the panel be given more detailed labels to indicate sex.

3. L352-358 appears to have a half sentence that is hard to follow

4. L297: change “had no affectation of” to “did not affect” and “glucose management” to “glucose homeostasis”

6. PLOS authors have the option to publish the peer review history of their article (what does this mean? ). If published, this will include your full peer review and any attached files.

**Do you want your identity to be public for this peer review?** For information about this choice, including consent withdrawal, please see our Privacy Policy .

Reviewer #1: No

Reviewer #2: No

---

## [Author Response · Author response to Decision Letter 0]

7 Mar 2025

We would like to thank the reviewers for their comments and detailed review of the manuscript. Their feedback has been helpful in enhancing our work. We are now submitting a revised version of the manuscript, which we believe addresses all the points raised by the reviewers. In this revised version, we have corrected all minor issues identified. Additionally, we have performed new experiments to validate and further strengthen the transcriptomic data that further support our original data. Please see below for responses to the reviewers, including corrections, comments, an improved discussion. These changes enhance the clarity and rigor of the manuscript.

Thank you for considering our revised manuscript for publication in Plos One..

Point by point response to the reviewer's comments:

Reviewer #1:

The authors present their findings from characterization of mature adipocyte specific knockout of Dbc1. Unexpectedly, no difference is observed in wildtype and cAT-dbc1 KO animals on normal chow or HFD except for enhanced immune signaling in KO adipocytes. The findings though new and important in context to the data on the whole body knockout mice, can be distilled into a short report rather than a primary research article. Most of the data on cAT-dbc1 KO mouse generation can be supplemental. RNA seq data on major immune genes should be validated by qPCR.

Response: We agree with the reviewer that RNA seq data on major immune genes as well as the original RNA seq data set should be validated as a whole, by qPCR. Therefore, we conducted qPCR analysis of several of the genes found involved in the reported pathways. Thus, we completed the study by including: a) genes not directly related to the main inflammatory pathway here reported, but found either significantly up or down regulated; b) genes belonging to canonical inflammatory pathways whose expression levels were found unchanged in our original RNAseq analysis. As expected, the gene expression levels found in the qPCR analysis closely correlated with those present in the seq data set. Importantly, because the strategy used for the transcriptome analysis (Gene Set Enrichment Analysis, GSEA), privileges the occurrence of minor but general changes in given metabolic pathways instead high changes in a minor set of genes to assign significant changes, qPCR analysis yielded subtle but consistent changes in several genes belonging to the reported inflammatory via. This information is now included in a supplementary figure (figure S5).

Reviewer: Methods must clearly show the number of mice used in each experiment. There are some minor errors of grammar, line formatting, etc.

Response: The number of mice used in each experiment is now clearly stated in the Methods section. Grammar error and line formatting were corrected.

Reviewer #2

Reviewer: The transcription factor Deleted in Breast Cancer 1 (DBC1) has previously been shown to regulate obesity and aging-related processes in preclinical models, with murine globalDbc1 KO causing morbid obesity but protecting against liver steatosis, insulin resistance and atherosclerosis. This makes DBC1 a promising therapeutic target. In this study the authors generated an adipocyte-specific Dbc1 KO model to assess whether the putatively beneficial effects of global Dbc1 KO are attributable to adipocyte Dbc1 KO. Deletion of Dbc1 in mature adipocytes had no effect on weight gain, glucose tolerance nor other markers of metabolic dysregulation, regardless of sex, however. This is the central finding of this study, and although it is essentially a “negative” finding this is a clear and valuable answer to the question posed. Dbc1 KO adipocytes were, however, reported to display an mRNA expression profile consistent with increased inflammation, although the magnitude of this was not very impressive, and the finding was not pursued in more detail. In summary, this study adds new and useful information regarding the metabolic role of adipocyte Dbc1. In several places the drafting of the manuscript could also be included.

Major Comments

The manuscript states that “In previous studies, we and others showed that whole-body Dbc1knockout mice do not develop obesity when fed a regular chow diet” (L291-293) but later states that “Our previous work, in mice with mixed genetic background, showed that whole-body Dbc1 KO mice develop morbid obesity with prevention of fatty-acid spillover and metabolic damage” (L314-316). Can this be reconciled? Presumably it relates to diet?

Response: As pointed out by the reviewer, the effect of morbid but “healthy” obesity developed by Dbc1 KO mice only occurs when mice are fed with high-fat diet. This apparent inconsistency arises because we omitted to clearly state that animals that get obese but without fatty acid spillover and metabolic disorders were those fed a high fat diet.…The corrected sentence now state “… whole-body Dbc1 KO mice fed with high fat diet for 20 weeks, develop morbid obesity with prevention of fatty-acid spillover and metabolic damage.” (L344-346)

We thank the reviewer for pointing out this error.

Reviewer: Fig 1: In Panel D, top, the immunoblot shows a clear decrease of Dbdc1 protein expression in Cre+ whole eWAT and iWAT and corresponding isolated adipocytes, suggesting successful deletion of Dbc1 in mature adipocytes. However, expression of Dbdc1 in isolated SVF seems increased compared to related whole fat tissue, particularly in the last lane, possibly indicating compensatory upregulation in SV cells from Cre+ fat tissue. Can the authors quantify the blot and clarify this?

Response: We quantified the expression of Dbc1 in the SVF by western blot and found that there is no increase in Dbc1 expression in isolated SVF of Cre+ mice. This fact is now briefly mentioned at the end of the section “Generation of Dbc1LoxP/LoxP conditional mice by CRISPR/Cas9 and adipocyte-specific deletion” (L316-319) and depicted in figure S1. We thank the reviewer for pointing out this important detail.

Reviewer: Dbc1 knockout (KO) adipocytes show upregulation of several inflammatory pathways (Fig 6D). Particularly noticeable are the enriched terms related to cytokine production and NF-kB activity. This may indicate the adipose tissue is inflamed in KO mice, but this requires substantial further characterisation if much is to be made of this observation. As noted above, the negative metabolic findings are perhaps the most important outcome of the study.

Response: We agree about the need for a deeper characterization of these findings. To evaluate the consistency of the original RNA seq data as a whole, we have now included qPCR analysis to confirm the expression of several key genes belonging to differentially regulated inflammatory pathways. Also, we completed the study by including: a) some additional genes not directly related to the main inflammatory pathway (but found either up or down regulated), b) genes belonging to canonical inflammatory pathways whose expression levels were found unchanged in our original RNAseq dataset. As a result, we found that the expression levels of selected genes found in the qPCR analysis, closely correlated with most of those present in the seq data set. This information is now included in a supplementary figure (figure S5).

Minor comments

Reviewer: The layouts of figure legends, and panel labelling are not very clear. As an example, in L267-289, the illustration jumps back and forward or repeats description of the same things.

Response: We revised panel labeling and figure legends. Some text passages were rewritten as well as some figures rearranged. The changes are highlighted in red.

Reviewer: Nomenclature for the mouse strain is sometimes inconsistent: cAT-Dbc1 KO vs Dbcloxp/loxp Cre +/- (L291- 301). Also, can the panel be given more detailed labels to indicate sex.

Response: We replaced “cAT-Dbc1 KO” with “Dbc1LoxP/LoxP;CRE” throughout the text to designate the mouse adipocyte conditional Dbc1 KO strain.

Reviewer: L352-358 appears to have a half sentence that is hard to follow.

Response: Response: The text was reviewed and rewritten for clarity (L384-388). The revised paragraph is as follows:

"Histological analysis of liver samples confirmed the presence of steatosis, as expected in the diet-induced obesity (DIO) condition (Fig. 5). Overall, Dbc1LoxP/LoxP;CRE KO mice did not show significant differences in weight gain, glucose management, or liver and kidney function compared to controls under a high-fat diet treatment, indicating that adipocyte-specific Dbc1 deletion does not replicate the protective effects observed in whole-body Dbc1 knockout models."

Reviewer: L297: change “had no affectation of” to “did not affect” and “glucose management” to “glucose homeostasis”

Response: We accepted and incorporated these suggestions.

---

## [Decision Letter · Decision Letter 1]

27 Mar 2025

Adipocyte-specific deletion of Dbc1 does not recapitulate healthy obesity phenotype but suggests regulation of inflammation signaling

PONE-D-24-40711R1

Dear Dr. Escande,

We’re pleased to inform you that your manuscript has been judged scientifically suitable for publication and will be formally accepted for publication once it meets all outstanding technical requirements.

Kind regards,

Chien-Feng Li, M.D., Ph.D.

Academic Editor

PLOS ONE

Additional Editor Comments (optional):

Reviewers' comments:

Reviewer's Responses to Questions

**Comments to the Author**

1. If the authors have adequately addressed your comments raised in a previous round of review and you feel that this manuscript is now acceptable for publication, you may indicate that here to bypass the “Comments to the Author” section, enter your conflict of interest statement in the “Confidential to Editor” section, and submit your "Accept" recommendation.

Reviewer #1: All comments have been addressed

Reviewer #3: All comments have been addressed

2. Is the manuscript technically sound, and do the data support the conclusions?

Reviewer #1: Yes

Reviewer #3: Yes

3. Has the statistical analysis been performed appropriately and rigorously? 

Reviewer #1: Yes

Reviewer #3: Yes

4. Have the authors made all data underlying the findings in their manuscript fully available?

Reviewer #1: Yes

Reviewer #3: No

5. Is the manuscript presented in an intelligible fashion and written in standard English?

Reviewer #1: Yes

Reviewer #3: Yes

6. Review Comments to the Author

Reviewer #1: (No Response)

Reviewer #3: (No Response)

7. PLOS authors have the option to publish the peer review history of their article (what does this mean? ). If published, this will include your full peer review and any attached files.

**Do you want your identity to be public for this peer review?** For information about this choice, including consent withdrawal, please see our Privacy Policy .

Reviewer #1: No

Reviewer #3: No

---

## [Editor Report · Acceptance letter]

PONE-D-24-40711R1

PLOS ONE

Dear Dr. Escande,

I'm pleased to inform you that your manuscript has been deemed suitable for publication in PLOS ONE. Congratulations! Your manuscript is now being handed over to our production team.

Kind regards,

on behalf of

Dr. Chien-Feng Li

Academic Editor

PLOS ONE